# Two-Layer Feature Reduction for Sparse-Group Lasso via Decomposition of Convex Sets

**Jie Wang, Jieping Ye**
Computer Science and Engineering
Arizona State University, Tempe, AZ 85287
{jie.wang.ustc, jieping.ye}@asu.edu

## Abstract

Sparse-Group Lasso (SGL) has been shown to be a powerful regression technique for simultaneously discovering group and within-group sparse patterns by using a combination of the $\ell_1$ and $\ell_2$ norms. However, in large-scale applications, the complexity of the regularizers entails great computational challenges. In this paper, we propose a novel **t**wo-**l**ayer **f**eature **re**duction method (TLFre) for SGL via a decomposition of its dual feasible set. The two-layer reduction is able to quickly identify the inactive groups and the inactive features, respectively, which are guaranteed to be absent from the sparse representation and can be removed from the optimization. Existing feature reduction methods are only applicable for sparse models with one sparsity-inducing regularizer. To our best knowledge, TLFre is *the first one* that is capable of dealing with *multiple* sparsity-inducing regularizers. Moreover, TLFre has a very low computational cost and can be integrated with any existing solvers. Experiments on both synthetic and real data sets show that TLFre improves the efficiency of SGL by orders of magnitude.

## 1 Introduction

Sparse-Group Lasso (SGL) [5, 16] is a powerful regression technique in identifying important groups and features simultaneously. To yield sparsity at both group and individual feature levels, SGL combines the Lasso [18] and group Lasso [28] penalties. In recent years, SGL has found great success in a wide range of applications, including but not limited to machine learning [20, 27], signal processing [17], bioinformatics [14] etc. Many research efforts have been devoted to developing efficient solvers for SGL [5, 16, 10, 21]. However, when the feature dimension is extremely high, the complexity of the SGL regularizers imposes great computational challenges. Therefore, there is an increasingly urgent need for nontraditional techniques to address the challenges posed by the massive volume of the data sources.

Recently, El Ghaoui et al. [4] proposed a promising feature reduction method, called *SAFE screening*, to screen out the so-called *inactive* features, which have zero coefficients in the solution, from the optimization. Thus, the size of the data matrix needed for the training phase can be significantly reduced, which may lead to substantial improvement in the efficiency of solving sparse models. Inspired by SAFE, various exact and heuristic feature screening methods have been proposed for many sparse models such as Lasso [25, 11, 19, 26], group Lasso [25, 22, 19], etc. It is worthwhile to mention that the discarded features by exact feature screening methods such as SAFE [4], DOME [26] and EDPP [25] are guaranteed to have zero coefficients in the solution. However, heuristic feature screening methods like Strong Rule [19] may mistakenly discard features which have nonzero coefficients in the solution. More recently, the idea of exact feature screening has been extended to exact sample screening, which screens out the nonsupport vectors in SVM [13, 23] and LAD [23]. As a promising data reduction tool, exact feature/sample screening would be of great practical importance because they can effectively reduce the data size without sacrificing the optimality [12].

However, all of the existing feature/sample screening methods are only applicable for the sparse models with one sparsity-inducing regularizer. In this paper, we propose an exact two-layer feature screening method, called TLFre, for the SGL problem. The two-layer reduction is able to quickly identify the inactive groups and the inactive features, respectively, which are guaranteed to have zero coefficients in the solution. To the best of our knowledge, TLFre is the first screening method which is capable of dealing with multiple sparsity-inducing regularizers.

We note that most of the existing exact feature screening methods involve an estimation of the dual optimal solution. The difficulty in developing screening methods for sparse models with multiple sparsity-inducing regularizers like SGL is that the dual feasible set is the sum of simple convex sets. Thus, to determine the feasibility of a given point, we need to know if it is decomposable with respect to the summands, which is itself a nontrivial problem (see Section 2). One of our major contributions is that we derive an elegant decomposition method of any dual feasible solutions of SGL via the framework of Fenchel's duality (see Section 3). Based on the Fenchel's dual problem of SGL, we motivate TLFre by an in-depth exploration of its geometric properties and the optimality conditions. We derive the set of the regularization parameter values corresponding to zero solutions. To develop TLFre, we need to estimate the upper bounds involving the dual optimal solution. To this end, we first give an accurate estimation of the dual optimal solution via the normal cones. Then, we formulate the estimation of the upper bounds via nonconvex optimization problems. We show that these nonconvex problems admit closed form solutions. Experiments on both synthetic and real data sets demonstrate that the speedup gained by TLFre in solving SGL can be orders of magnitude. All proofs are provided in the long version of this paper [24].

**Notation**: Let $\|\cdot\|_1$, $\|\cdot\|$ and $\|\cdot\|_\infty$ be the $\ell_1$, $\ell_2$ and $\ell_\infty$ norms, respectively. Denote by $\mathcal{B}_1^n$, $\mathcal{B}^n$, and $\mathcal{B}_\infty^n$ the unit $\ell_1$, $\ell_2$, and $\ell_\infty$ norm balls in $\mathbb{R}^n$ (we omit the superscript if it is clear from the context). For a set $\mathcal{C}$, let $\text{int}\,\mathcal{C}$ be its interior. If $\mathcal{C}$ is closed and convex, we define the projection operator as $\mathbf{P}_\mathcal{C}(\mathbf{w}) := \text{argmin}_{\mathbf{u}\in\mathcal{C}} \|\mathbf{w}-\mathbf{u}\|$. We denote by $\mathbf{I}_\mathcal{C}(\cdot)$ the indicator function of $\mathcal{C}$, which is 0 on $\mathcal{C}$ and $\infty$ elsewhere. Let $\Gamma_0(\mathbb{R}^n)$ be the class of proper closed convex functions on $\mathbb{R}^n$. For $f \in \Gamma_0(\mathbb{R}^n)$, let $\partial f$ be its subdifferential. The domain of $f$ is the set $\text{dom}\, f := \{\mathbf{w} : f(\mathbf{w}) < \infty\}$. For $\mathbf{w} \in \mathbb{R}^n$, let $[\mathbf{w}]_i$ be its $i^{th}$ component. For $\gamma \in \mathbb{R}$, let $\text{sgn}(\gamma) = \text{sign}(\gamma)$ if $\gamma \neq 0$, and $\text{sgn}(0) = 0$. We define

$$\text{SGN}(\mathbf{w}) = \left\{ \mathbf{s} \in \mathbb{R}^n : [\mathbf{s}]_i \in \begin{cases} \text{sign}([\mathbf{w}]_i), & \text{if } [\mathbf{w}]_i \neq 0; \\ [-1,1], & \text{if } [\mathbf{w}]_i = 0. \end{cases} \right\}$$

We denote by $\gamma_+ = \max(\gamma, 0)$. Then, the shrinkage operator $\mathcal{S}_\gamma(\mathbf{w}) : \mathbb{R}^n \to \mathbb{R}^n$ with $\gamma \geq 0$ is

$$[\mathcal{S}_\gamma(\mathbf{w})]_i = (|[\mathbf{w}]_i| - \gamma)_+ \text{sgn}([\mathbf{w}]_i),\ i = 1,\ldots,n. \tag{1}$$

## 2 Basics and Motivation

In this section, we briefly review some basics of SGL. Let $\mathbf{y} \in \mathbb{R}^N$ be the response vector and $\mathbf{X} \in \mathbb{R}^{N\times p}$ be the matrix of features. With the group information available, the SGL problem [5] is

$$\min_{\beta\in\mathbb{R}^p} \frac{1}{2}\left\|\mathbf{y} - \sum\nolimits_{g=1}^{G}\mathbf{X}_g\beta_g\right\|^2 + \lambda_1\sum\nolimits_{g=1}^{G}\sqrt{n_g}\|\beta_g\| + \lambda_2\|\beta\|_1, \tag{2}$$

where $n_g$ is the number of features in the $g^{th}$ group, $\mathbf{X}_g \in \mathbb{R}^{N\times n_g}$ denotes the predictors in that group with the corresponding coefficient vector $\beta_g$, and $\lambda_1, \lambda_2$ are positive regularization parameters. Without loss of generality, let $\lambda_1 = \alpha\lambda$ and $\lambda_2 = \lambda$ with $\alpha > 0$. Then, problem (2) becomes:

$$\min_{\beta\in\mathbb{R}^p} \frac{1}{2}\left\|\mathbf{y} - \sum\nolimits_{g=1}^{G}\mathbf{X}_g\beta_g\right\|^2 + \lambda\left(\alpha\sum\nolimits_{g=1}^{G}\sqrt{n_g}\|\beta_g\| + \|\beta\|_1\right). \tag{3}$$

By the Lagrangian multipliers method [24], the dual problem of SGL is

$$\sup_\theta \left\{ \frac{1}{2}\|\mathbf{y}\|^2 - \frac{1}{2}\left\|\frac{\mathbf{y}}{\lambda} - \theta\right\|^2 : \mathbf{X}_g^T\theta \in \mathcal{D}_g^\alpha := \alpha\sqrt{n_g}\mathcal{B} + \mathcal{B}_\infty,\ g = 1,\ldots,G \right\}. \tag{4}$$

It is well-known that the dual feasible set of Lasso is the intersection of closed half spaces (thus a polytope); for group Lasso, the dual feasible set is the intersection of ellipsoids. Surprisingly, the geometric properties of these dual feasible sets play fundamentally important roles in most of the existing screening methods for sparse models with one sparsity-inducing regularizer [23, 11, 25, 4].

When we incorporate multiple sparse-inducing regularizers to the sparse models, problem (4) indicates that the dual feasible set can be much more complicated. Although (4) provides a geometric

description of the dual feasible set of SGL, it is not suitable for further analysis. Notice that, *even the feasibility of a given point $\theta$ is not easy to determine*, since it is nontrivial to tell if $\mathbf{X}_g^T\theta$ can be decomposed into $\mathbf{b}_1 + \mathbf{b}_2$ with $\mathbf{b}_1 \in \alpha\sqrt{n_g}\mathcal{B}$ and $\mathbf{b}_2 \in \mathcal{B}_\infty$. Therefore, to develop screening methods for SGL, it is desirable to gain deeper understanding of the sum of simple convex sets.

In the next section, we analyze the dual feasible set of SGL in depth via the Fenchel's Duality Theorem. We show that for each $\mathbf{X}_g^T\theta \in \mathcal{D}_g^\alpha$, Fenchel's duality naturally leads to an explicit decomposition $\mathbf{X}_g^T\theta = \mathbf{b}_1 + \mathbf{b}_2$, with one belonging to $\alpha\sqrt{n_g}\mathcal{B}$ and the other one belonging to $\mathcal{B}_\infty$. This lays the foundation of the proposed screening method for SGL.

# 3 The Fenchel's Dual Problem of SGL

In Section 3.1, we derive the Fenchel's dual of SGL via Fenchel's Duality Theorem. We then motivate TLFre and sketch our approach in Section 3.2. In Section 3.3, we discuss the geometric properties of the Fenchel's dual of SGL and derive the set of $(\lambda, \alpha)$ leading to zero solutions.

## 3.1 The Fenchel's Dual of SGL via Fenchel's Duality Theorem

To derive the Fenchel's dual problem of SGL, we need the Fenchel's Duality Theorem as stated in Theorem 1. The conjugate of $f \in \Gamma_0(\mathbb{R}^n)$ is the function $f^* \in \Gamma_0(\mathbb{R}^n)$ defined by

$$f^*(\mathbf{z}) = \sup_{\mathbf{w}} \langle \mathbf{w}, \mathbf{z} \rangle - f(\mathbf{w}).$$

**Theorem 1.** [Fenchel's Duality Theorem] *Let $f \in \Gamma_0(\mathbb{R}^N)$, $\Omega \in \Gamma_0(\mathbb{R}^p)$, and $\mathcal{T}(\beta) = \mathbf{y} - \mathbf{X}\beta$ be an affine mapping from $\mathbb{R}^p$ to $\mathbb{R}^N$. Let $p^*, d^* \in [-\infty, \infty]$ be primal and dual values defined, respectively, by the Fenchel problems:*

$$p^* = \inf_{\beta \in \mathbb{R}^p} f(\mathbf{y} - \mathbf{X}\beta) + \lambda\Omega(\beta); \quad d^* = \sup_{\theta \in \mathbb{R}^N} -f^*(\lambda\theta) - \lambda\Omega^*(\mathbf{X}^T\theta) + \lambda\langle \mathbf{y}, \theta \rangle.$$

*One has $p^* \geq d^*$. If, furthermore, $f$ and $\Omega$ satisfy the condition $0 \in \text{int}\,(\text{dom}\,f - \mathbf{y} + \mathbf{X}\text{dom}\,\Omega)$, then the equality holds, i.e., $p^* = d^*$, and the supreme is attained in the dual problem if finite.*

We omit the proof of Theorem 1 since it is a slight modification of Theorem 3.3.5 in [2].

Let $f(\mathbf{w}) = \frac{1}{2}\|\mathbf{w}\|^2$, and $\lambda\Omega(\beta)$ be the second term in (3). Then, SGL can be written as

$$\min_\beta\ f(\mathbf{y} - \mathbf{X}\beta) + \lambda\Omega(\beta).$$

To derive the Fenchel's dual problem of SGL, Theorem 1 implies that we need to find $f^*$ and $\Omega^*$. It is well-known that $f^*(\mathbf{z}) = \frac{1}{2}\|\mathbf{z}\|^2$. Therefore, we only need to find $\Omega^*$, where the concept *infimal convolution* is needed. Let $h, g \in \Gamma_0(\mathbb{R}^n)$. The infimal convolution of $h$ and $g$ is defined by

$$(h\square g)(\xi) = \inf_\eta\ h(\eta) + g(\xi - \eta),$$

and it is exact at a point $\xi$ if there exists a $\eta^*(\xi)$ such that $(h\square g)(\xi) = h(\eta^*(\xi)) + g(\xi - \eta^*(\xi))$. $h\square g$ is exact if it is exact at every point of its domain, in which case it is denoted by $h\boxdot g$.

**Lemma 2.** *Let $\Omega_1^\alpha(\beta) = \alpha\sum_{g=1}^G \sqrt{n_g}\|\beta_g\|$, $\Omega_2(\beta) = \|\beta\|_1$ and $\Omega(\beta) = \Omega_1^\alpha(\beta) + \Omega_2(\beta)$. Moreover, let $\mathcal{C}_g^\alpha = \alpha\sqrt{n_g}\mathcal{B} \subset \mathbb{R}^{n_g}$, $g = 1, \ldots, G$. Then, the following hold:*

(i): $(\Omega_1^\alpha)^*(\xi) = \sum_{g=1}^G \mathbf{I}_{\mathcal{C}_g^\alpha}(\xi_g)$, $\quad$ $(\Omega_2)^*(\xi) = \sum_{g=1}^G \mathbf{I}_{\mathcal{B}_\infty}(\xi_g)$,

(ii): $\Omega^*(\xi) = ((\Omega_1^\alpha)^* \boxdot (\Omega_2)^*)(\xi) = \sum_{g=1}^G \mathbf{I}_{\mathcal{B}}\left(\frac{\xi_g - \mathbf{P}_{\mathcal{B}_\infty}(\xi_g)}{\alpha\sqrt{n_g}}\right)$,

*where $\xi_g \in \mathbb{R}^{n_g}$ is the sub-vector of $\xi$ corresponding to the $g^{th}$ group.*

Note that $\mathbf{P}_{\mathcal{B}_\infty}(\xi_g)$ admits a closed form solution, i.e., $[\mathbf{P}_{\mathcal{B}_\infty}(\xi_g)]_i = \text{sgn}\,([\xi_g]_i)\min\,(|[\xi_g]_i|, 1)$. Combining Theorem 1 and Lemma 2, the Fenchel's dual of SGL can be derived as follows.

**Theorem 3.** *For the SGL problem in (3), the following hold:*

(i): *The Fenchel's dual of SGL is given by:*

$$\inf_\theta \left\{ \tfrac{1}{2}\|\tfrac{\mathbf{y}}{\lambda} - \theta\|^2 - \tfrac{1}{2}\|\mathbf{y}\|^2 : \left\|\mathbf{X}_g^T\theta - \mathbf{P}_{\mathcal{B}_\infty}(\mathbf{X}_g^T\theta)\right\| \leq \alpha\sqrt{n_g},\ g = 1, \ldots, G \right\}. \tag{5}$$

(ii): *Let $\beta^*(\lambda, \alpha)$ and $\theta^*(\lambda, \alpha)$ be the optimal solutions of problems (3) and (5), respectively. Then,*

$$\lambda\theta^*(\lambda, \alpha) = \mathbf{y} - \mathbf{X}\beta^*(\lambda, \alpha), \tag{6}$$

$$\mathbf{X}_g^T\theta^*(\lambda, \alpha) \in \alpha\sqrt{n_g}\partial\|\beta_g^*(\lambda, \alpha)\| + \partial\|\beta_g^*(\lambda, \alpha)\|_1,\ g = 1, \ldots, G. \tag{7}$$

**Remark 1.** *We note that the shrinkage operator can also be expressed by*

$$\mathcal{S}_\gamma(\mathbf{w}) = \mathbf{w} - \mathbf{P}_{\gamma\mathcal{B}_\infty}(\mathbf{w}), \;\; \gamma \geq 0. \tag{8}$$

*Therefore, problem (5) can be written more compactly as*

$$\inf_\theta \left\{ \tfrac{1}{2}\|\tfrac{\mathbf{y}}{\lambda} - \theta\|^2 - \tfrac{1}{2}\|\mathbf{y}\|^2 : \left\|\mathcal{S}_1(\mathbf{X}_g^T\theta)\right\| \leq \alpha\sqrt{n_g}, \, g = 1, \ldots, G \right\}. \tag{9}$$

**Remark 2.** *Eq. (6) and Eq. (7) can be obtained by the Fenchel-Young inequality [2, 24]. They are the so-called KKT conditions [3] and can also be obtained by the Lagrangian multiplier method [24]. Moreover, for the SGL problem, its Lagrangian dual in (4) and Fenchel's dual in (5) are indeed equivalent to each other [24].*

**Remark 3.** *An appealing advantage of the Fenchel's dual in (5) is that we have a natural decomposition of all points $\xi_g \in \mathcal{D}_g^\alpha$: $\xi_g = \mathbf{P}_{\mathcal{B}_\infty}(\xi_g) + \mathcal{S}_1(\xi_g))$ with $\mathbf{P}_{\mathcal{B}_\infty}(\xi_g) \in \mathcal{B}_\infty$ and $\mathcal{S}_1(\xi_g) \in \mathcal{C}_g^\alpha$. As a result, this leads to a convenient way to determine the feasibility of any dual variable $\theta$ by checking if $\mathcal{S}_1(\mathbf{X}_g^T\theta) \in \mathcal{C}_g^\alpha$, $g = 1, \ldots, G$.*

### 3.2 Motivation of the Two-Layer Screening Rules

We motive the two-layer screening rules via the KKT condition in Eq. (7). As implied by the name, there are two layers in our method. The first layer aims to identify the inactive groups, and the second layer is designed to detect the inactive features for the remaining groups.

by Eq. (7), we have the following cases by noting $\partial\|\mathbf{w}\|_1 = \mathrm{SGN}(\mathbf{w})$ and

$$\partial\|\mathbf{w}\| = \begin{cases} \left\{ \frac{\mathbf{w}}{\|\mathbf{w}\|} \right\}, & \text{if } \mathbf{w} \neq 0, \\ \{\mathbf{u} : \|\mathbf{u}\| \leq 1\}, & \text{if } \mathbf{w} = 0. \end{cases}$$

**Case 1.** If $\beta_g^*(\lambda, \alpha) \neq 0$, we have

$$[\mathbf{X}_g^T\theta^*(\lambda,\alpha)]_i \in \begin{cases} \alpha\sqrt{n_g}\frac{[\beta_g^*(\lambda,\alpha)]_i}{\|\beta_g^*(\lambda,\alpha)\|} + \mathrm{sign}([\beta_g^*(\lambda,\alpha)]_i), & \text{if } [\beta_g^*(\lambda,\alpha)]_i \neq 0, \\ [-1,1], & \text{if } [\beta_g^*(\lambda,\alpha)]_i = 0. \end{cases} \tag{10}$$

In view of Eq. (10), we can see that

$$\text{(a): } \mathcal{S}_1(\mathbf{X}_g^T\theta^*(\lambda,\alpha)) = \alpha\sqrt{n_g}\frac{\beta_g^*(\lambda_1,\lambda_2)}{\|\beta_g^*(\lambda_1,\lambda_2)\|} \text{ and } \|\mathcal{S}_1(\mathbf{X}_g^T\theta^*(\lambda,\alpha))\| = \alpha\sqrt{n_g}, \tag{11}$$

$$\text{(b): If } \left|[\mathbf{X}_g^T\theta^*(\lambda,\alpha]_i\right| \leq 1 \text{ then } [\beta_g^*(\lambda,\alpha)]_i = 0. \tag{12}$$

**Case 2.** If $\beta_g^*(\lambda, \alpha) = 0$, we have

$$[\mathbf{X}_g^T\theta^*(\lambda,\alpha)]_i \in \alpha\sqrt{n_g}[\mathbf{u}_g]_i + [-1,1], \|\mathbf{u}_g\| \leq 1. \tag{13}$$

**The first layer (group-level) of TLFre** From (11) in **Case 1**, we have

$$\left\|\mathcal{S}_1(\mathbf{X}_g^T\theta^*(\lambda,\alpha))\right\| < \alpha\sqrt{n_g} \Rightarrow \beta_g^*(\lambda,\alpha) = 0. \tag{R1}$$

Clearly, (R1) can be used to identify the inactive groups and thus a group-level screening rule.

**The second layer (feature-level) of TLFre** Let $\mathbf{x}_{g_i}$ be the $i^{th}$ column of $\mathbf{X}_g$. We have $[\mathbf{X}_g^T\theta^*(\lambda,\alpha)]_i = \mathbf{x}_{g_i}^T\theta^*(\lambda,\alpha)$. In view of (12) and (13), we can see that

$$\left|\mathbf{x}_{g_i}^T\theta^*(\lambda,\alpha)\right| \leq 1 \Rightarrow [\beta_g^*(\lambda,\alpha)]_i = 0. \tag{R2}$$

Different from (R1), (R2) detects the inactive features and thus it is a feature-level screening rule.

However, we cannot directly apply (R1) and (R2) to identify the inactive groups/features because both need to know $\theta^*(\lambda,\alpha)$. Inspired by the SAFE rules [4], we can first estimate a region $\Theta$ containing $\theta^*(\lambda,\alpha)$. Let $\mathbf{X}_g^T\Theta = \{\mathbf{X}_g^T\theta : \theta \in \Theta\}$. Then, (R1) and (R2) can be relaxed as follows:

$$\sup_{\xi_g} \left\{ \|\mathcal{S}_1(\xi_g)\| : \xi_g \in \Xi_g \supseteq \mathbf{X}_g^T\Theta \right\} < \alpha\sqrt{n_g} \Rightarrow \beta_g^*(\lambda,\alpha) = 0, \tag{R1*}$$

$$\sup_\theta \left\{ \left|\mathbf{x}_{g_i}^T\theta\right| : \theta \in \Theta \right\} \leq 1 \Rightarrow [\beta_g^*(\lambda,\alpha)]_i = 0. \tag{R2*}$$

Inspired by (R1*) and (R2*), we develop TLFre via the following three steps:
**Step 1.** Given $\lambda$ and $\alpha$, we estimate a region $\Theta$ that contains $\theta^*(\lambda,\alpha)$.
**Step 2.** We solve for the supreme values in (R1*) and (R2*).
**Step 3.** By plugging in the supreme values from **Step 2**, (R1*) and (R2*) result in the desired two-layer screening rules for SGL.

### 3.3 The Set of Parameter Values Leading to Zero Solution

For notational convenience, let $\mathcal{F}_g^\alpha = \{\theta : \|\mathcal{S}_1(\mathbf{X}_g^T\theta)\| \leq \alpha\sqrt{n_g}\}$, $g = 1, \ldots, G$; and thus the feasible set of the Fenchel's dual of SGL is $\mathcal{F}^\alpha = \cap_{g=1,\ldots,G} \mathcal{F}_g^\alpha$. In view of problem (5) [or (9)], we can see that $\theta^*(\lambda, \alpha)$ is the projection of $\mathbf{y}/\lambda$ on $\mathcal{F}^\alpha$, i.e., $\theta^*(\lambda, \alpha) = \mathbf{P}_{\mathcal{F}^\alpha}(\mathbf{y}/\lambda)$. Thus, if $\mathbf{y}/\lambda \in \mathcal{F}^\alpha$, we have $\theta^*(\lambda, \alpha) = \mathbf{y}/\lambda$. Moreover, by (R1), we can see that $\beta^*(\lambda, \alpha) = 0$ if $\mathbf{y}/\lambda$ is an *interior* point of $\mathcal{F}^\alpha$. Indeed, we have the following stronger result.

**Theorem 4.** *For the SGL problem, let $\lambda_{\max}^\alpha = \max_g \{\rho_g : \|\mathcal{S}_1(\mathbf{X}_g^T\mathbf{y}/\rho_g)\| = \alpha\sqrt{n_g}\}$. Then,*
$$\tfrac{\mathbf{y}}{\lambda} \in \mathcal{F}^\alpha \Leftrightarrow \theta^*(\lambda, \alpha) = \tfrac{\mathbf{y}}{\lambda} \Leftrightarrow \beta^*(\lambda, \alpha) = 0 \Leftrightarrow \lambda \geq \lambda_{\max}^\alpha.$$

$\rho_g$ in the definition of $\lambda_{\max}^\alpha$ admits a closed form solution [24]. Theorem 4 implies that the optimal solution $\beta^*(\lambda, \alpha)$ is 0 as long as $\mathbf{y}/\lambda \in \mathcal{F}^\alpha$. This geometric property also leads to an explicit characterization of the set of $(\lambda_1, \lambda_2)$ such that the corresponding solution of problem (2) is 0. We denote by $\bar{\beta}^*(\lambda_1, \lambda_2)$ the optimal solution of problem (2).

**Corollary 5.** *For the SGL problem in (2), let $\lambda_1^{\max}(\lambda_2) = \max_g \frac{1}{\sqrt{n_g}} \|\mathcal{S}_{\lambda_2}(\mathbf{X}_g^T\mathbf{y})\|$. Then,*

(i): $\bar{\beta}^*(\lambda_1, \lambda_2) = 0 \Leftrightarrow \lambda_1 \geq \lambda_1^{\max}(\lambda_2)$.

(ii): *If $\lambda_1 \geq \lambda_1^{\max} := \max_g \frac{1}{\sqrt{n_g}} \|\mathbf{X}_g^T\mathbf{y}\|$ or $\lambda_2 \geq \lambda_2^{\max} := \|\mathbf{X}^T\mathbf{y}\|_\infty$, then $\bar{\beta}^*(\lambda_1, \lambda_2) = 0$.*

## 4 The Two-Layer Screening Rules for SGL

We follow the three steps in Section 3.2 to develop TLFre. In Section 4.1, we give an accurate estimation of $\theta^*(\lambda, \alpha)$ via normal cones [15]. Then, we compute the supreme values in (R1$^*$) and (R2$^*$) by solving nonconvex problems in Section 4.2. We present the TLFre rules in Section 4.3.

### 4.1 Estimation of the Dual Optimal Solution

Because of the geometric property of the dual problem in (5), i.e., $\theta^*(\lambda, \alpha) = \mathbf{P}_{\mathcal{F}^\alpha}(\mathbf{y}/\lambda)$, we have a very useful characterization of the dual optimal solution via the so-called normal cones [15].

**Definition 1.** [15] *For a closed convex set $\mathcal{C} \in \mathbb{R}^n$ and a point $\mathbf{w} \in \mathcal{C}$, the normal cone to $\mathcal{C}$ at $\mathbf{w}$ is*
$$N_\mathcal{C}(\mathbf{w}) = \{\mathbf{v} : \langle \mathbf{v}, \mathbf{w}' - \mathbf{w} \rangle \leq 0, \forall \mathbf{w}' \in \mathcal{C}\}. \tag{14}$$

By Theorem 4, $\theta^*(\bar{\lambda}, \alpha)$ is known if $\bar{\lambda} = \lambda_{\max}^\alpha$. Thus, we can estimate $\theta^*(\lambda, \alpha)$ in terms of $\theta^*(\bar{\lambda}, \alpha)$. Due to the same reason, *we only consider the cases with $\lambda < \lambda_{\max}^\alpha$ for $\theta^*(\lambda, \alpha)$ to be estimated.*

**Remark 4.** *In many applications, the parameter values that perform the best are usually unknown. To determine appropriate parameter values, commonly used approaches such as cross validation and stability selection involve solving SGL many times over a grip of parameter values. Thus, given $\{\alpha^{(i)}\}_{i=1}^{\mathcal{I}}$ and $\lambda^{(1)} \geq \cdots \geq \lambda^{(\mathcal{J})}$, we can fix the value of $\alpha$ each time and solve SGL by varying the value of $\lambda$. We repeat the process until we solve SGL for all of the parameter values.*

**Theorem 6.** *For the SGL problem in (3), suppose that $\theta^*(\bar{\lambda}, \alpha)$ is known with $\bar{\lambda} \leq \lambda_{\max}^\alpha$. Let $\rho_g$, $g = 1, \ldots, G$, be defined by Theorem 4. For any $\lambda \in (0, \bar{\lambda})$, we define*
$$\mathbf{n}_\alpha(\bar{\lambda}) = \begin{cases} \mathbf{y}/\bar{\lambda} - \theta^*(\bar{\lambda}, \alpha), & \text{if } \bar{\lambda} < \lambda_{\max}^\alpha \\ \mathbf{X}_*\mathcal{S}_1(\mathbf{X}_*^T\mathbf{y}/\lambda_{\max}^\alpha), & \text{if } \bar{\lambda} = \lambda_{\max}^\alpha, \end{cases} \text{ where } \mathbf{X}_* = \operatorname{argmax}_{\mathbf{X}_g} \rho_g,$$
$$\mathbf{v}_\alpha(\lambda, \bar{\lambda}) = \tfrac{\mathbf{y}}{\lambda} - \theta^*(\bar{\lambda}, \alpha),$$
$$\mathbf{v}_\alpha(\lambda, \bar{\lambda})^\perp = \mathbf{v}_\alpha(\lambda, \bar{\lambda}) - \tfrac{\langle \mathbf{v}_\alpha(\lambda, \bar{\lambda}), \mathbf{n}_\alpha(\bar{\lambda}) \rangle}{\|\mathbf{n}_\alpha(\bar{\lambda})\|^2} \mathbf{n}_\alpha(\bar{\lambda}).$$

*Then, the following hold:*

(i): $\mathbf{n}_\alpha(\bar{\lambda}) \in N_{\mathcal{F}^\alpha}(\theta^*(\bar{\lambda}, \alpha))$,

(ii): $\|\theta^*(\lambda, \alpha) - (\theta^*(\bar{\lambda}, \alpha) + \frac{1}{2}\mathbf{v}_\alpha^\perp(\lambda, \bar{\lambda}))\| \leq \frac{1}{2}\|\mathbf{v}_\alpha^\perp(\lambda, \bar{\lambda})\|$.

For notational convenience, let $\mathbf{o}_\alpha(\lambda, \bar{\lambda}) = \theta^*(\bar{\lambda}, \alpha) + \frac{1}{2}\mathbf{v}_\alpha^\perp(\lambda, \bar{\lambda})$. Theorem 6 shows that $\theta^*(\lambda, \alpha)$ lies inside the ball of radius $\frac{1}{2}\|\mathbf{v}_\alpha^\perp(\lambda, \bar{\lambda})\|$ centered at $\mathbf{o}_\alpha(\lambda, \bar{\lambda})$.

### 4.2 Solving for the supreme values via Nonconvex Optimization

We solve the optimization problems in (R1$^*$) and (R2$^*$). To simplify notations, let
$$\Theta = \{\theta : \|\theta - \mathbf{o}_\alpha(\lambda, \bar{\lambda})\| \leq \tfrac{1}{2}\|\mathbf{v}_\alpha^\perp(\lambda, \bar{\lambda})\|\}, \tag{15}$$
$$\Xi_g = \{\xi_g : \|\xi_g - \mathbf{X}_g^T\mathbf{o}_\alpha(\lambda, \bar{\lambda})\| \leq \tfrac{1}{2}\|\mathbf{v}_\alpha^\perp(\lambda, \bar{\lambda})\|\|\mathbf{X}_g\|_2\}, \ g = 1, \ldots, G. \tag{16}$$

Theorem 6 indicates that $\theta^*(\lambda, \alpha) \in \Theta$. Moreover, we can see that $\mathbf{X}_g^T \Theta \subseteq \Xi_g$, $g = 1, \ldots, G$. To develop the TLFre rule by (R1*) and (R2*), we need to solve the following optimization problems:

$$s_g^*(\lambda, \bar{\lambda}; \alpha) = \sup_{\xi_g} \{\|\mathcal{S}_1(\xi_g)\| : \xi_g \in \Xi_g\}, \; g = 1, \ldots, G, \tag{17}$$

$$t_{g_i}^*(\lambda, \bar{\lambda}; \alpha) = \sup_{\theta} \{|\mathbf{x}_{g_i}^T \theta| : \theta \in \Theta\}, \; i = 1, \ldots, n_g, \; g = 1, \ldots, G. \tag{18}$$

**Solving problem (17)** We consider the following equivalent problem of (17):

$$\tfrac{1}{2}\left(s_g^*(\lambda, \bar{\lambda}; \alpha)\right)^2 = \sup_{\xi_g} \left\{\tfrac{1}{2}\|\mathcal{S}_1(\xi_g)\|^2 : \xi_g \in \Xi_g\right\}. \tag{19}$$

We can see that the objective function of problem (19) is *continuously differentiable* and the feasible set is a ball. Thus, (19) is a *nonconvex* problem because we need to *maximize* a convex function subject to a convex set. We derive the closed form solutions of problems (17) and (19) as follows.

**Theorem 7.** *For problems (17) and (19), let* $\mathbf{c} = \mathbf{X}_g^T \mathbf{o}_\alpha(\lambda, \bar{\lambda})$, $r = \tfrac{1}{2}\|\mathbf{v}_\alpha^\perp(\lambda, \bar{\lambda})\|\|\mathbf{X}_g\|_2$ *and* $\Xi_g^*$ *be the set of the optimal solutions.*
(i) *Suppose that* $\mathbf{c} \notin \mathcal{B}_\infty$, *i.e.,* $\|\mathbf{c}\|_\infty > 1$. *Let* $\mathbf{u} = r\mathcal{S}_1(\mathbf{c})/\|\mathcal{S}_1(\mathbf{c})\|$. *Then,*

$$s_g^*(\lambda, \bar{\lambda}; \alpha) = \|\mathcal{S}_1(\mathbf{c})\| + r \quad \text{and} \quad \Xi_g^* = \{\mathbf{c} + \mathbf{u}\}. \tag{20}$$

(ii) *Suppose that* $\mathbf{c}$ *is a boundary point of* $\mathcal{B}_\infty$, *i.e.,* $\|\mathbf{c}\|_\infty = 1$. *Then,*

$$s_g^*(\lambda, \bar{\lambda}; \alpha) = r \quad \text{and} \quad \Xi_g^* = \{\mathbf{c} + \mathbf{u} : \mathbf{u} \in N_{\mathcal{B}_\infty}(\mathbf{c}), \|\mathbf{u}\| = r\}. \tag{21}$$

(iii) *Suppose that* $\mathbf{c} \in \text{int} \, \mathcal{B}_\infty$, *i.e.,* $\|\mathbf{c}\|_\infty < 1$. *Let* $i^* \in \mathcal{I}^* = \{i : |[\mathbf{c}]_i| = \|\mathbf{c}\|_\infty\}$. *Then,*

$$s_g^*(\lambda, \bar{\lambda}; \alpha) = (\|\mathbf{c}\|_\infty + r - 1)_+, \tag{22}$$

$$\Xi_g^* = \begin{cases} \Xi_g, & \text{if } \Xi_g \subset \mathcal{B}_\infty, \\ \{\mathbf{c} + r \cdot \text{sgn}([\mathbf{c}]_{i^*})\mathbf{e}_{i^*} : i^* \in \mathcal{I}^*\}, & \text{if } \Xi_g \not\subset \mathcal{B}_\infty \text{ and } \mathbf{c} \neq 0, \\ \{r \cdot \mathbf{e}_{i^*}, -r \cdot \mathbf{e}_{i^*} : i^* \in \mathcal{I}^*\}, & \text{if } \Xi_g \not\subset \mathcal{B}_\infty \text{ and } \mathbf{c} = 0, \end{cases}$$

where $\mathbf{e}_i$ is the $i^{th}$ standard basis vector.

**Solving problem in (18)** Problem (18) can be solved directly via the Cauchy-Schwarz inequality.
**Theorem 8.** *For problem (18), we have* $t_{g_i}^*(\lambda, \bar{\lambda}; \alpha) = |\mathbf{x}_{g_i}^T \mathbf{o}_\alpha(\lambda, \bar{\lambda})| + \tfrac{1}{2}\|\mathbf{v}_\alpha^\perp(\lambda, \bar{\lambda})\|\|\mathbf{x}_{g_i}\|$.

### 4.3  The Proposed Two-Layer Screening Rules

To develop the two-layer screening rules for SGL, we only need to plug the supreme values $s_g^*(\lambda_2, \bar{\lambda}_2; \lambda_1)$ and $t_{g_i}^*(\lambda_2, \bar{\lambda}_2; \lambda_1)$ in (R1*) and (R2*). We present the TLFre rule as follows.
**Theorem 9.** *For the SGL problem in (3), suppose that we are given* $\alpha$ *and a sequence of parameter values* $\lambda_{\max}^\alpha = \lambda^{(0)} > \lambda^{(1)} > \ldots > \lambda^{(\mathcal{J})}$. *For each integer* $0 \leq j < \mathcal{J}$, *we assume that* $\beta^*(\lambda^{(j)}, \alpha)$ *is known. Let* $\theta^*(\lambda^{(j)}, \alpha)$, $\mathbf{v}_\alpha^\perp(\lambda^{(j+1)}, \lambda^{(j)})$ *and* $s_g^*(\lambda^{(j+1)}, \lambda^{(j)}; \alpha)$ *be given by Eq. (6), Theorems 6 and 7, respectively. Then, for* $g = 1, \ldots, G$, *the following holds*

$$s_g^*(\lambda^{(j+1)}, \lambda^{(j)}; \alpha) < \alpha\sqrt{n_g} \Rightarrow \beta_g^*(\lambda^{(j+1)}, \alpha) = 0. \tag{$\mathcal{L}_1$}$$

*For the* $\hat{g}^{th}$ *group that does not pass the rule in ($\mathcal{L}_1$), we have* $[\beta_{\hat{g}}^*(\lambda^{(j+1)}, \alpha)]_i = 0$ *if*

$$\left|\mathbf{x}_{\hat{g}_i}^T\left(\frac{\mathbf{y} - \mathbf{X}\beta^*(\lambda^{(j)}, \alpha)}{\lambda^{(j)}} + \tfrac{1}{2}\mathbf{v}_\alpha^\perp(\lambda^{(j+1)}, \lambda^{(j)})\right)\right| + \tfrac{1}{2}\|\mathbf{v}_\alpha^\perp(\lambda^{(j+1)}, \lambda^{(j)})\|\|\mathbf{x}_{\hat{g}_i}\| \leq 1. \tag{$\mathcal{L}_2$}$$

($\mathcal{L}_1$) and ($\mathcal{L}_2$) are the first layer and second layer screening rules of TLFre, respectively.

## 5  Experiments

We evaluate TLFre on both synthetic and real data sets. To measure the performance of TLFre, we compute the *rejection ratios* of ($\mathcal{L}_1$) and ($\mathcal{L}_2$), respectively. Specifically, let $m$ be the number of features that have 0 coefficients in the solution, $\overline{\mathcal{G}}$ be the index set of groups that are discarded by ($\mathcal{L}_1$) and $\overline{p}$ be the number of inactive features that are detected by ($\mathcal{L}_2$). The rejection ratios of ($\mathcal{L}_1$) and ($\mathcal{L}_2$) are defined by $r_1 = \frac{\sum_{g \in \overline{\mathcal{G}}} n_g}{m}$ and $r_2 = \frac{|\overline{p}|}{m}$, respectively. Moreover, we report the *speedup* gained by TLFre, i.e., the ratio of the running time of solver without screening to the running time of solver with TLFre. The solver used in this paper is from SLEP [9].

To determine appropriate values of $\alpha$ and $\lambda$ by cross validation or stability selection, we can run TLFre with as many parameter values as we need. Given a data set, for illustrative purposes only, we select seven values of $\alpha$ from $\{\tan(\psi) : \psi = 5°, 15°, 30°, 45°, 60°, 75°, 85°\}$. Then, for each value of $\alpha$, we run TLFre along a sequence of 100 values of $\lambda$ equally spaced on the logarithmic scale of $\lambda/\lambda_{\max}^\alpha$ from 1 to 0.01. Thus, 700 pairs of parameter values of $(\lambda, \alpha)$ are sampled in total.

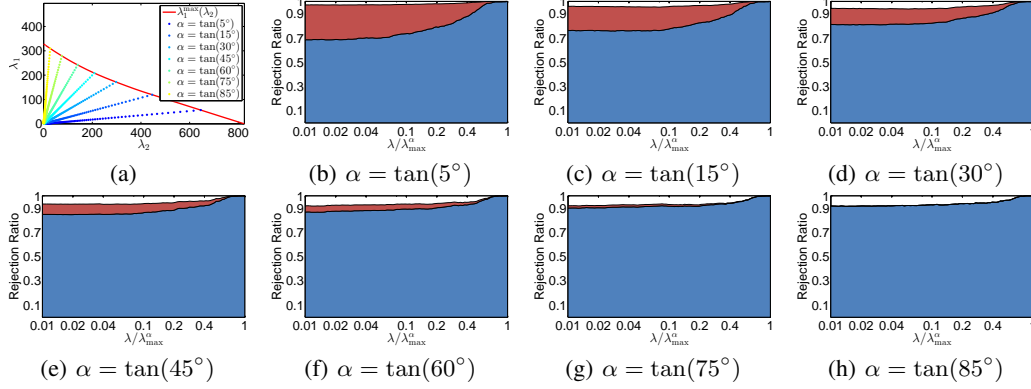

Figure 1: Rejection ratios of TLFre on the Synthetic 1 data set.

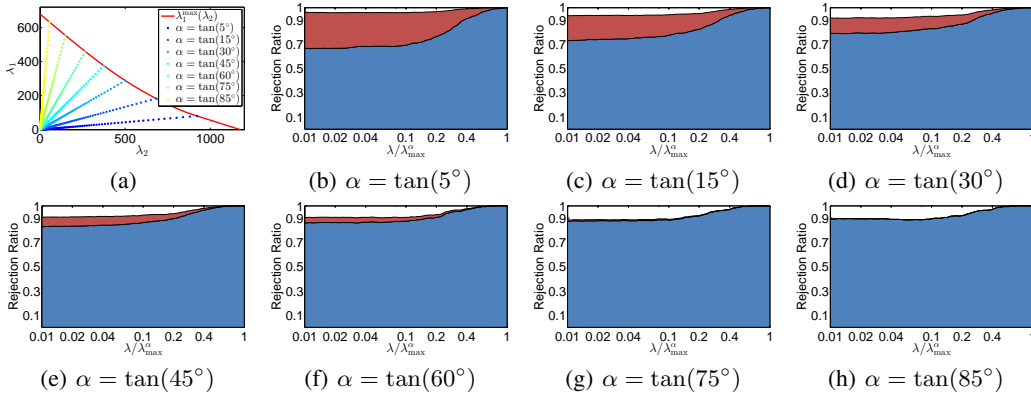

Figure 2: Rejection ratios of TLFre on the Synthetic 2 data set.

## 5.1 Simulation Studies

We perform experiments on two synthetic data sets that are commonly used in the literature [19, 29]. The true model is $\mathbf{y} = \mathbf{X}\beta^* + 0.01\epsilon$, $\epsilon \sim N(0,1)$. We generate two data sets with $250 \times 10000$ entries: Synthetic 1 and Synthetic 2. We randomly break the 10000 features into 1000 groups. For Synthetic 1, the entries of the data matrix $\mathbf{X}$ are i.i.d. standard Gaussian with pairwise correlation zero, i.e., $\mathrm{corr}(\mathbf{x}_i, \mathbf{x}_i) = 0$. For Synthetic 2, the entries of the data matrix $\mathbf{X}$ are drawn from i.i.d. standard Gaussian with pairwise correlation $0.5^{|i-j|}$, i.e., $\mathrm{corr}(\mathbf{x}_i, \mathbf{x}_j) = 0.5^{|i-j|}$. To construct $\beta^*$, we first randomly select $\gamma_1$ percent of groups. Then, for each selected group, we randomly select $\gamma_2$ percent of features. The selected components of $\beta^*$ are populated from a standard Gaussian and the remaining ones are set to 0. We set $\gamma_1 = \gamma_2 = 10$ for Synthetic 1 and $\gamma_1 = \gamma_2 = 20$ for Synthetic 2.

The figures in the upper left corner of Fig. 1 and Fig. 2 show the plots of $\lambda_1^{\max}(\lambda_2)$ (see Corollary 5) and the sampled parameter values of $\lambda$ and $\alpha$ (recall that $\lambda_1 = \alpha\lambda$ and $\lambda_2 = \lambda$). For the other figures, the blue and red regions represent the rejection ratios of $(\mathcal{L}_1)$ and $(\mathcal{L}_2)$, respectively. We can see that TLFre is very effective in discarding inactive groups/features; that is, more than $90\%$ of inactive features can be detected. Moreover, we can observe that the first layer screening $(\mathcal{L}_1)$ becomes more effective with a larger $\alpha$. Intuitively, this is because the group Lasso penalty plays a more important role in enforcing the sparsity with a larger value of $\alpha$ (recall that $\lambda_1 = \alpha\lambda$). The top and middle parts of Table 1 indicate that the speedup gained by TLFre is very significant (up to 30 times) and TLFre is very efficient. Compared to the running time of the solver without screening, the running time of TLFre is negligible. The running time of TLFre includes that of computing $\|\mathbf{X}_g\|_2$, $g = 1, \ldots, G$, which can be efficiently computed by the power method [6]. Indeed, this can be shared for TLFre with different parameter values.

## 5.2 Experiments on Real Data Set

We perform experiments on the Alzheimer's Disease Neuroimaging Initiative (ADNI) data set (http://adni.loni.usc.edu/). The data matrix consists of $747$ samples with $426040$ single

Table 1: Running time (in seconds) for solving SGL along a sequence of $100$ tuning parameter values of $\lambda$ equally spaced on the logarithmic scale of $\lambda/\lambda_{\max}^{\alpha}$ from $1.0$ to $0.01$ by (a): the solver [9] without screening; (b): the solver combined with TLFre. The top and middle parts report the results of TLFre on Synthetic 1 and Synthetic 2. The bottom part reports the results of TLFre on the ADNI data set with the GMV data as response.

| | $\alpha$ | $\tan(5^\circ)$ | $\tan(15^\circ)$ | $\tan(30^\circ)$ | $\tan(45^\circ)$ | $\tan(60^\circ)$ | $\tan(75^\circ)$ | $\tan(85^\circ)$ |
|---|---|---|---|---|---|---|---|---|
| | solver | 298.36 | 301.74 | 308.69 | 307.71 | 311.33 | 307.53 | 291.24 |
| Synthetic 1 | TLFre | 0.77 | 0.78 | 0.79 | 0.79 | 0.81 | 0.79 | 0.77 |
| | TLFre+solver | 10.26 | 12.47 | 15.73 | 17.69 | 19.71 | 21.95 | 22.53 |
| | **speedup** | **29.09** | **24.19** | **19.63** | **17.40** | 15.79 | **14.01** | **12.93** |
| | solver | 294.64 | 294.92 | 297.29 | 297.50 | 297.59 | 295.51 | 292.24 |
| Synthetic 2 | TLFre | 0.79 | 0.80 | 0.80 | 0.81 | 0.81 | 0.81 | 0.82 |
| | TLFre+solver | 11.05 | 12.89 | 16.08 | 18.90 | 20.45 | 21.58 | 22.80 |
| | **speedup** | **26.66** | **22.88** | **18.49** | **15.74** | **14.55** | **13.69** | **12.82** |
| | solver | 30652.56 | 30755.63 | 30838.29 | 31096.10 | 30850.78 | 30728.27 | 30572.35 |
| ADNI+GMV | TLFre | 64.08 | 64.56 | 64.96 | 65.00 | 64.89 | 65.17 | 65.05 |
| | TLFre+solver | 372.04 | 383.17 | 386.80 | 402.72 | 391.63 | 385.98 | 382.62 |
| | **speedup** | **82.39** | **80.27** | **79.73** | **77.22** | **78.78** | **79.61** | **79.90** |

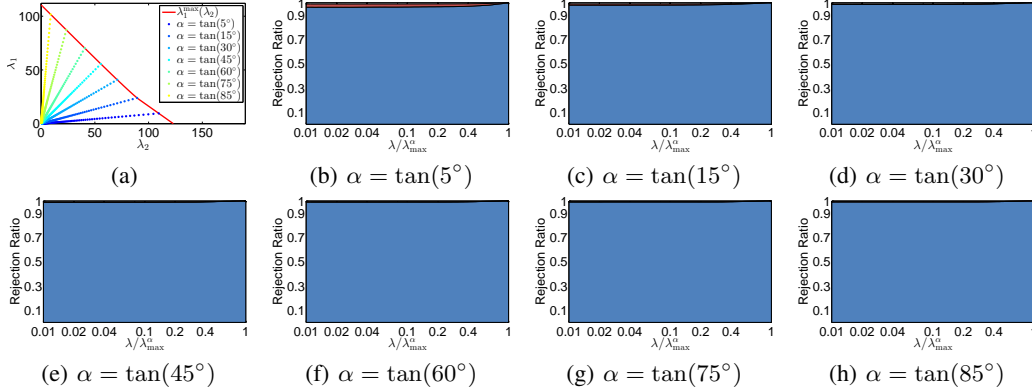

Figure 3: Rejection ratios of TLFre on the ADNI data set with grey matter volume as response.

nucleotide polymorphisms (SNPs), which are divided into $94765$ groups. The response vector is the grey matter volume (GMV).

The figure in the upper left corner of Fig. 3 shows the plots of $\lambda_1^{\max}(\lambda_2)$ (see Corollary 5) and the sampled parameter values of $\alpha$ and $\lambda$. The other figures present the rejection ratios of $(\mathcal{L}_1)$ and $(\mathcal{L}_2)$ by blue and red regions, respectively. We can see that almost all of the inactive groups/features are discarded by TLFre. The rejection ratios of $r_1 + r_2$ are very close to $1$ in all cases. The bottom part of Table 1 shows that TLFre leads to a very significant speedup (about $80$ times). In other words, the solver without screening needs about eight and a half hours to solve the $100$ SGL problems for each value of $\alpha$. However, combined with TLFre, the solver needs only six to eight minutes. Moreover, we can observe that the computational cost of TLFre is negligible compared to that of the solver without screening. This demonstrates the efficiency of TLFre.

## 6  Conclusion

In this paper, we propose a novel feature reduction method for SGL via decomposition of convex sets. We also derive the set of parameter values that lead to zero solutions of SGL. To the best of our knowledge, TLFre is the first method which is applicable to sparse models with multiple sparsity-inducing regularizers. More importantly, the proposed approach provides novel framework for developing screening methods for complex sparse models with multiple sparsity-inducing regularizers, e.g., $\ell_1$ SVM that performs both sample and feature selection, fused Lasso and tree Lasso with more than two regularizers. Experiments on both synthetic and real data sets demonstrate the effectiveness and efficiency of TLFre. We plan to generalize the idea of TLFre to $\ell_1$ SVM, fused Lasso and tree Lasso, which are expected to consist of multiple layers of screening.

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
