[Reviews · NeurIPS 2014]

Submitted by Assigned_Reviewer_24

The authors give a safe screening rule for the sparse group lasso. It characterizes the nonzero features in terms of the dual parameters. By bounding the maximum change of the dual parameters as a function of the change in lambda they can efficiently and "safely" screen out variables guaranteed to stay zero. These ideas are related to SAFE rules for the lasso: a method which gives a univariate screen based on inner-products which can "safely" determine candidate non-zero features in the Lasso.

Quality: The paper is of high quality. The authors build these ideas from the ground up (though the framework is based on similar "Safe rules" for the lasso). The idea is useful, and the mathematics are fleshed out. In addition, the strength of the method is demonstrated on reasonable examples. My only complaint is that the paper is a bit symbol/notation dense, and I think that some of the details could have been swapped to a supplement (and some additional intuition might have been added), though the supplement was already quite long and dense.

Clarity: The math is clearly explained. Again, it is a bit dense, but with a 9 page limit this is understandable.

Originality: Safe rules have been used before in Lasso problems. Nonetheless this is a tricky extension and required substantial non-obvious work to put together.

Significance: This is a very useful method. The sparse group lasso has become a relatively standard tool, and this screen can really help with the computational tractability of large-scale problems.
Summary: This is a strong computational tool for screening features in the sparse group lasso. It's clearly explained and quite useful.

Submitted by Assigned_Reviewer_44

This paper presented an approach for studying convex functions that would recover sparsity at both the group and individual level using L1 and L2 regularizers. Numerical experiments were presented that include both real and simulated data sets, which showcased the efficiency of the proposed approach. Additional theoretical motivations for the approach, and derivations of the parameters that would lead to zero solutions were presented.

I thought that this paper was a very interesting study on a great topic. It presents an approach that is of interest to both practitioners, but has sufficient theory to provide insight into the chosen approach. I have not managed to reach the paper quite rigorously but I believe it to be well written and of good quality.
Summary: I believe this paper is appropriate for NIPS given the nice range of theory and practicality. It studies a problem that is of interest to the audience of NIPS, and is nicely written.

Submitted by Assigned_Reviewer_45

--- Summary ---

This paper generalizes the screening procedure to the sparse group lasso problem where it is able to identify inactive groups and inactive features (i.e., features with zero weight in the optimal solution) simultaneously before optimization and consequently to accomplish dimensionality reduction. Therefore, when used as a per-processing step in sparse group lasso problem to reduce its scale, the computational complexity of the optimization problem can be reduced significantly without sacrificing the accuracy of the model.

--- Contributions and overall evaluation ---

To apply the typical screening methods, one needs to estimate the optimal dual solution of the original optimization problem. For sparse group lasso, the feasible set is an intersection of ellipsoids and it turns out that the dual estimation is a difficult task. The basic idea to remedy this difficulty is to utilize Fenchel's duality to derive an elegant decomposition of the dual feasible set (i.e., sum of simple convex sets due to the pretense of multiple sparsity inducing regularizes) to be able to estimate the dual objective.

The paper does survey related work appropriately and is clearly written. The paper technically sound and the proofs seem to be correct. It is clear, of high quality, novel, definitely of significance. However, more effort can be put into making the proofs more understandable (though many of the explanations are very good) and extending the experiments.

-- Minor comments ---

- Line 181, Page 4, by --> By
- Line 527, Page 10, |\beta|_1 --> |\beta_g|_1

Summary: This paper proposes a two-layer screening procedure for sparse groups lasso problem. The solution is novel and the paper has a good quality with sound theoretical and empirical results.
Author Feedback
Author rebuttal: We thank all reviewers for the constructive comments.

Reviewer 24

Q: The paper is of high quality. My only complaint is that the paper is a bit symbol/notation dense, and I think that some of the details could have been swapped to a supplement (and some additional intuition might have been added), though the supplement was already quite long and dense.

A: Thanks for the compliment. We will add more details to the main text to improve the readability of the final version, if accepted.

Reviewer 44

Q: This paper is appropriate for NIPS given the nice range of theory and practicality. It is a very interesting study on a great topic, and is nicely written.

A: Thanks for the compliment.

Reviewer 45

Q: This paper is clear, of high quality, novel, definitely of significance. However, more effort can be put into making the proofs more understandable (though many of the explanations are very good) and extending the experiments.

A: Thanks for the compliment. We will further improve the readability of the final version, if accepted.

Q: There are some typos in line 181 and 527.

A: Thanks for pointing out those typos. We will correct all of them in the final version, if accepted.